# Nutrition Education Intervention Increases Fish Consumption among School Children in Indonesia: Results from Behavioral Based Randomized Control Trial

**DOI:** 10.3390/ijerph17196970

**Published:** 2020-09-23

**Authors:** Trias Mahmudiono, Triska Susila Nindya, Qonita Rachmah, Calista Segalita, Luh Ade Ari Wiradnyani

**Affiliations:** 1Department of Nutrition, Faculty of Public Health, Universitas Airlangga, Jl. Mulyorejo Kampus C, Surabaya 60115, Indonesia; triska.nindya@fkm.unair.ac.id (T.S.N.); qonita.rachmah@fkm.unair.ac.id (Q.R.); calsglt@gmail.com (C.S.); 2Southeast Asian Ministers of Education Organization Regional Centre for Food and Nutrition (SEAMEO RECFON) Pusat Kajian Gizi Regional (PKGR), Universitas Indonesia, Jl. Salemba Raya No.6, Jakarta Pusat 10430, Indonesia; awiradnyani@seameo-recfon.org

**Keywords:** nutrition education, fish consumption, school children, raised bed pool, Indonesia

## Abstract

This study aimed to analyze the effectiveness of behavioral-based nutrition education to increase fish consumption among school children using a raised bed pool. This was a randomized control trial study with a 3-months nutrition education intervention using a raised bed pool, as a medium to improve their internalization to increase fish consumption behavior. A paired *t*-test was used to calculate the difference in the increase of fish consumption, knowledge, attitude, perceived behavioral control, subjective norm, and intention. This study took place in a majority of low to medium urban households in Surabaya in Sidotopo Wetan I and Sidotopo Wetan II elementary school. Elementary school children at 4th and 5th grade and mother of elementary school children with 104 children were eligible and willing to participate. After the completion of interventions, significant improvement in delta-mean and effectiveness observed in attitude, subjective norm, perceived behavioral control, intention, knowledge, and fish consumption (*p* < 0.001). The 3 months of nutrition education intervention based on the theory of planned behavior significantly increase fish consumption among elementary school children. The increased consumption was believed to be related to the increase in children’s knowledge and attitude towards consuming fish.

## 1. Introduction

Hidden hunger, a micronutrient deficiency such as iron, iodine, vitamin A, or calcium, is one of the nutrition problems that still exist in a developing country, including Indonesia. The prevalence of anemia in schoolchildren globally reached 37% in which was found to be higher in Asian children than African [1]. Based on the Indonesia Family Life Survey (IFLS), the prevalence of anemia in children aged 5–12 years declined from 36.4% in 2000 to 20.6% in 2008 [2]. The latest report shows that the prevalence of anemia in children aged 5–14 years is 26.4% and count as a public health problem [3]. A study in one primary school in Surabaya found that the anemia prevalence in the schoolchild reached 13.2% [4].

In the short term, anemia in schoolchildren affects the level of learning concentration due to reduced oxygen supply to the brain causes a lack of hemoglobin [5]. In the long term, the condition of anemia in children causes several health consequences, especially during childhood development. Anemia was more at risk for developing neurologic delays [6].

A study in Makassar found that schoolchildren who only occasionally (2–3 times/week) consume heme protein sources are more at risk of anemia than those who frequently (4–7 times/week) consume [7]. Protein sources affect the type of iron that can be absorbed by the body and could lead to anemia if absorption is low.

Fish is one source of heme protein that has a good iron absorption rate. Protein content in fish reaches 18% and consists of essential amino acids. According to the Indonesian Food Exchange List, one serving of fresh fish (50 g) contains 10 g of protein and 2 mg of iron. The Total Diet Study in 2014 reported the average consumption of fish and processed fish meat. The children group (aged 5–12 years) was 70.7 g per person per day [8]. East Java is one of the provinces with the highest growth of fish consumption per kg/capita/year since 2010–2014. Fish consumption in 2010 was only 19.01 kg/capita/ year up to 27.89 kg/capita/year with a 46% growth level in the last 4 years. Fish consumption in East Java is predicted to continue to rise [9]. In 2019, fish consumption is targeted to reach 54.49 kg/capita/year [10].

Surabaya is included as the second largest metropolitan city in Indonesia, with approximately 2.8 million people [11]. With a large population, community health insurance based on Community Health Enterprises (SMEs) is perceived to be a practical option. Public health issues such as anemia require a comprehensive policy approach, although, in developing countries, resource limitations are often becoming a constraint [12]. However, given the magnitude of the impact of anemia on the quality of Indonesian human resources, efforts should be made to overcome the problem with the improvement of the intervention method. Nutrition education is one of the most cost-effective interventions and resulting in a long-lasting impact [13]. One of the behavior change theories that is widely used in nutrition education is the Theory of Planned Behavior (TPB) [14]. TPB addresses the potential for changes in participants’ “desire,” “intention,” “attitudes,” “perceived behavioral control,” and “skills” for achieving the sought-after behavioral outcome of increased fish consumption as drawn by Figure 1.

The intervention of youth garden program based on the Theory of Planned Behavior succeeded in improving the attitude of the children related to the increase of vegetables and fruits consumption both in boys (B = 1.525; *p*-value < 0.001) and girls (B = 1.421; *p*-value < 0.001), as well as increased of perceived behavior control in consuming vegetables and fruits in girls (B = 0.303; *p*-value = 0.014) [15]. Nutrition education interventions based on school gardens are widely used and succeeded in increasing the desire and intention to try eating vegetables and fruits in children, children’s knowledge of the importance of eating vegetables and fruits, as well as improving children’s attitudes and skills to increase vegetable and fruit consumption [16,17]. Another study also mentioned that school-based gardens are useful because of their experiential and direct learning in schoolchildren [18]. 

Analogous to involving children in school garden activities to increase consumption of fruits and vegetables, increased consumption of fish may be triggered by involving children in fish farming activities in the school pool (raised bed pool). Raised Bed Pool (RBP) can be made using simple materials such as tarpaulins and wood/bamboo buffers that are filled with water for aquaculture, so they do not have to dig the soil. During this time, RBP has been implemented in several regions in Indonesia through the NICE program in six provinces in Indonesia namely North Sumatra, South Sumatera, West Nusa Tenggara, East Nusa Tenggara, West Kalimantan, and South Sulawesi. However, the use of RBP is limited only as a source of animal protein and the potential to be used as nutrition education media is still not widely recognized. By using RBP media in nutrition education as an instant reminder for schoolchildren is expected to increase school children’s intention to eat fish. The RBP Project also targets variables to increase school children’s perceived behavior control to eat fish with a weekly fishmeal program, fish-game cards, and catfish-based food menu making. Besides, RBP will also conduct a “Catfish Hours” program where school children are asked to help provide food for catfish raised in the RBP program.

Likely, a nutritional education intervention program with RBP to improve the achievement of the central government program “Love Eating Fish” and the prevention of anemia in school children can follow the success of the school gardening program. School gardening and raised bed pool equally prioritize the concept of mastery experience for children to be actively involved in the management of gardening programs or cultivating freshwater fish. Compared to the raised bed garden, the raised bed pool program produces an animal food source of heme-iron protein that is relatively easily absorbed by the body than non-heme iron. School children’s characteristics are more interested in interacting with moving objects rather than stationary objects. Together with RBP, providing nutrition education to improve school children’s knowledge of the importance of eating fish for growth and prevention of anemia, then school children involved in the RBP program will be able to associate new knowledge information with their activities while taking care the RBP. Based on the description above, this research would like to evaluate the effectiveness of RBP as a medium of nutrition education intervention to increase fish consumption as part of anemia prevention efforts in school children. Our hypothesis that behavioral-based nutrition education using a “raised bed pool” could improve fish consumption of school children.

## 2. Materials and Methods 

### 2.1. Study Setting and Sample

The study was conducted at Sidotopo Wetan I and Sidotopo Wetan II elementary school, Sidotopo Sub-district, Surabaya, Indonesia. Sidotopo is a region in Surabaya, with the majority of the people, are having low to medium monthly income. This district is included as one of the slum districts in North Surabaya. According to Surabaya Regional Statistical Survey [11], Sidotopo had a higher migration rate compared to other subdistricts. Hence, with a narrow space to live and overloaded with migrants, the slum environment could not be hindered. Inclusion criteria for the study were school children aged 10–12 years, not allergic to fish, and not on a special diet due to health problems (i.e., type 1 diabetes diet, low protein diet, weight loss diet).

Meanwhile, the withdrawn criteria were for the subjects missing >50%, or more than 3, intervention sessions. The school was chosen purposively by considering the location of the Sidotopo Subdistrict has the majority of residence categorized in the low and middle income where protein (including fish) availability and consumption tends to be low. The selection of Sidotopo Wetan I and II elementary school was based on easy accessibility, there was no raised bed pool, and there was no similar research before. Research subjects were elementary school children at 4th and 5th grade and mother of elementary school children. Based on a preliminary survey conducted in both elementary schools, the total number of grade 4 and 5 students recorded were 900 children (research population). A screening then performed based on the inclusion criteria and 800 eligible subjects were retrieved. The sample size in this study was determined using a formula of sample size for comparing the mean of continuous measurement in two samples. Using a z-statistic to approximate the t-statistic with the effect size calculated from the results of McAleese et al. [19] on increased consumption of vegetables per serving in nutritional education interventions based on school gardening (*n* = 45, SD = 1.7, ES = 1.2) compared with the control group. By using 80% power and alpha of 0.05, the minimum samples obtained without cluster correction were 32 subjects for each group. Then, taking into account the design effect 1 + (ρ (m + 1) using cluster size 30 and Inter-cluster Correlation Coefficient (ICC) = 0.043 based on manual diabetic research [20]. The design effect formula accounted for *m* number of observations in each cluster and ρ (rho) is the intra-cluster correlation. The minimum sample required was 46 primary school children in each group with the consideration of 10% drop out, the sample in this study was 52 school children in each group. The total sample in this study was 104 school-age children aged between 10–12 years taken from eligible and willing participants. Matching was done to reduce the influence of bias due to education level and gender. Randomization of the sample study was performed using computer software to generate random numbers. The adapted CONSORT diagram in Figure 2 showed the sampling procedure of the study.

The independent variables in the study were the nutrition education intervention (utilizing RBP) and the control condition where children and mothers were given printed materials. The nutrition education intervention would be expected to influence the key outcome (dependent variable) of fish consumption by altering mediating variables (from the TPB), resulting in actual behavior change in the form of consumption of fish. The moderating variables in this study including characteristics of school children, socio-economic status (SES) of the schoolchild’s family, knowledge; mediating variables include attitudes, intentions, subjective norms, and perceived behavior control of school children to eat fish, while the behavioral outcome was fish consumption. Nutrition education would hypothetically result in more reported fish consumption than the use of printed materials alone. Fish consumption was measured using a food diary record in which students were trained to fill the record form before the baseline data were taken. A total of three-days food diary records on all food consumed were collected in a week based on the previous study mentioned that 3-day food records made the best agreement compared to another dietary assessment [19]. Before administered food, research assistants were trained to filling the food record form. After data collection, they also do the data recheck and input to the software. Dietary data were analyzed using food processor software *Nutrisurvey;* drawing from a database of Indonesian Food updated yearly by the Department of Nutrition, Universitas Airlangga, Indonesia.

### 2.2. Ethics

Ethics approval for this study was received from the Institutional Review Board (IRB) at the Faculty of Public Health Universitas Airlangga approved the trial (reference number: 159-KEPK) dated 26 April 2017. The ethics obtained were aligned with the Helsinki protocol ensuring animal welfare throughout the study. Align with the beneficence principle, the use of catfish in the study was beneficial towards acquiring new knowledge and evidence. Furthermore, the fish was commonly eaten in the site of the study hence benefitting the nutrient intake of the children. The Universal Trial Number (UTN) for this study is U1111-1199-992. This trial was registered in the Thai Clinical Trials Registry (TCTR) and was allocated trial registration number TCTR20171207002. This research project had been approved for registration at TCTR since 2017-12-04 12:01:33. Before the study began, the mother’s or child guardian was first called by the teachers to the parent’s gathering at school. The research leader then explaining the study details to the prospective subjects about the child’s enrollment. During recruitment, potential participants and their mothers were given verbal and written information about the study, and at least one week to think about participating. Verbal and written informed consent was obtained during the monthly community health post-meeting. Participants are free to withdraw from the study at any time without negative consequences.

### 2.3. Anthropometric Measurement

The child’s weight was measured using the Omron HBP-317 digital scale with a 0.01 kg correction and measured in light clothing without shoes. Height was measured to the nearest 0.1 cm using a stadiometer (SECA 213). Both weight and height were measured twice to ensure the result’s validity. The third measurement will be taken if the difference between two prior two measurements differs by more than 1%. Another measurement to assess the nutritional status of the children was body composition, including body fat and body muscle percentage, as well as the resting metabolic rate. These indicators were measured using Omron HBF-317 that validated with the SECA digital weight scale. Bodyweight and composition data were collected to understand the distribution of school children’s nutritional status, but it is not in terms of the impact of fish consumption. Rather, the assessment is to describe the school children by their nutritional status which is indicated by their growth indicators.

### 2.4. Characteristics and Nutrition Knowledge

A general questionnaire was developed to obtained parent’s social-economic status (SES) data that includes parent’s educational background, employment, and the number of a family member, literacy level, family income, and food expenses. A child’s nutritional knowledge was measured using a questionnaire that was previously validated [16]. The questionnaire consists of three parts; the first part focus on the nutrition and health knowledge with a total of six questions and the second part focuses on household serving size to measure how well children know about the portion, and the last part consists of questions related to MyPlate Indonesia. The household serving size questionnaire consists of six questions, while MyPlate Indonesia consists of an essay question in which children were asked to categorize several food items to their group.

### 2.5. Theory of Planned Behavior

Outcome related psychological data obtained in this study, including children’s attitude, subjective norm, perceived behavioral control, behavior, and intention toward fish consumption. All of the psychological data questionnaires were developed as Likert scale answers based on Bandura’s guide for constructing attitude, subjective norm, perceived behavioral control, behavior, and intention scales [20]. Children’s attitude to consuming fish as the source of animal protein was measured using a three-item questionnaire (e.g., attitude toward increasing food consumption, attitude towards consuming fish at least twice a week, and attitude toward consuming fish every day). Similar to attitude measurement, three other indices including perceived behavioral control (e.g., consuming food is easy, willingness to consume fish, and willingness to consume fish if they know the possible danger of not consuming fish). Intention to consume fish was also measured using three questions (e.g., intention to consume fish, trying to consume fish, and planning to consume fish). Additionally, subjective norms towards fish consumption measured using 6 indicators in eight questions, including behavioral belief strength, outcome evaluation, injunctive normative belief strength, motivation to compliance, descriptive normative belief.

### 2.6. Intervention

This was a randomized control trial (RCT) with intervention in the form of nutrition education for 3 months targeting school children. All school children in this study received nutrition education material (booklet) describing strategies to increase fish consumption according to the Theory of Planned Behavior (TPB) constructions. Following random allocation, children were then divided into 2 groups; control group (CON) and intervention group (RBP). The control group (CON) did not receive nutrition education or fish pool in their school environment but received sets of printed educational materials, including infographics, comics, and recipe books. We did not further contact with the CON participants following the delivery of printed materials. The intervention group (RBP) got 6 nutrition education sessions focusing on recommendations for fish favors, fish pond maintenance, and bento making practices with fish dishes, and eating fish together in school. The comic describes the benefit of catfish in increasing student’s concentration during school hours. Besides, the infographic described anemia risk factors, strategies to improve anemia condition, and nutritional values of fish. Fish that acquired in one RBP pool was farmed catfish as much as 5000 seeds in one raised bed pool of 2 × 3 m. The pool was build in the schoolyard provided with a rooftop for shade. At the end of the study, some of the fish were harvested and some were continued to be raised by school guards/personnel.

Moreover, stationeries provided each participant after consenting to involve the whole study; the same goes for the control group. Six education sessions for the RBP group will be given once every two weeks for three months. Each education season approximately takes 30–90 min depending on the kind of activity given which consist of the class session, interactive games, and making a goal setting. During the nutrition education sessions, hands-on activities were provided to help children improve their self-efficacy toward fish consumption. 

The nutrition education sessions were administered by three investigators in Indonesia with expertise in community nutrition from Nutrition Department, Public Health Faculty Universitas Airlangga, two investigators, hold a master’s degree in nutrition and one investigator holds a doctoral degree in nutrition. Six trained research assistants who each hold a bachelor’s degree in public health nutrition delivered a hands-on activity session.

### 2.7. Statistical Analyses

A paired *t*-test was used to analyze the difference in outcomes of control and intervention groups. This statistical analysis has been adjusted for possible confounders such as school children characteristics, SES, and household characteristics. All data analyses were performed at IBM SPSS Statistics 22.

## 3. Results

At the beginning of the study, the total number of participants was 104 consisting of 52 children aged 9–12 years in each intervention and comparison group. However, two children in the intervention group were dropped as a result of not completing all of the six educational sessions. The drop-out rate from the intervention group was 3.8%. Thus, the analyzed result of the rest of the 102 samples was able to illustrate the effectiveness of nutrition education intervention using a raised bed pool. According to the data in Table 1, the average primary school children age in this study was 11.5 and their average height was around 145 cm. The average weight of the control group was slightly higher (43.76 kg) compared to the intervention group (39.6 kg). However, based on the *t*-test independent analytical test, it was found that between the control and intervention group there was no significant difference in children’s characteristics in terms of age, height, total fat, and resting metabolic.

Unfortunately, 2 children in the control group were not completing the household characteristics but they were able to complete the study hence considered missing data. Hence we included in the final analysis of the study, even though as seen in Table 2, only 50 children in each group were presented for their household characteristics. As can be seen in Table 2, most of the respondent families were the nuclear family (80%), literate mother (92%), and senior high school graduates (46%). Most of the respondent mothers were housewives (85%) and also most of the respondent fathers were senior high school graduates (85%). Most of the family income was between IDR 500,000.00 to IDR 2,000,000.00. The average family income and outcome in the intervention group were slightly higher compared to the control group. However, there was no meaningful difference based on the Chi-Square result (*p*-value = 0.054). Compared to the minimum wage for the Surabaya district (IDR 3,296,212.50), most of the family income in our study was much lower [11].

After nutrition education was given for 3 months consisted of 6 sessions education, it was shown that there was a significant improvement in children’s attitude towards fish consumption score. As can be seen in Table 3, significantly changed children’s attitude toward fish consumption was found in the intervention group. The behavior itself was categorized into some categories such as children’s attitude towards fish consumption minimum twice a week was fun (*p*-value < 0.001), everyday fish consumption was beneficial for health (*p*-value = 0.001), fun (*p*-value = 0.003), and considered as a good habit (*p*-value = 0.001).

Table 4 showed the diverse average score of children’s subjective norm associated with fish consumption suggestion. Some of the children’s subjective norm in the control group decreased (delta mean negative) compared to the intervention group who had an increasing tendency (delta mean positive). However, the *t*-test paired result showed no significant result in increasing and decreasing the average score of children’s subjective norm in fish consumption in the control and intervention group.

Table 5 showed the diverse average score of children perceived behavioral control associated with fish consumption suggestion. Some of the children’s perceived behavior control indicators in the control group decreased (delta mean negative) compared to the intervention group who had increasing perceived behavior control (delta mean positive). Nonetheless, the *t*-test paired test did not show a significant difference in increasing and decreasing the average score of children perceived behavioral control in fish consumption in the control and intervention group.

As can be seen in Table 6, the result of nutrition education using a raised bed pool in this study did not contribute to a significant change for children’s intention to consume fish.

Children’s intention to consume fish increased in the intervention group with raised bed pool (∆mean = 0.02) meanwhile, in the control group the intention score decreased compared when it was in baseline and end-line study (∆mean = 0.15).

One of the important discoveries in this study was that there was a significant improvement in the number of fish consumed by students in the intervention group (*p*-value = 0.022) after 6 education sessions and the use of a raised bed pool. Meanwhile, in the control group, there was no difference in the number of fish consumed in a day after three months of the study had been done (*p*-value = 0.184). In the intervention group, the number of fish consumption in a day was too small therefore the increasing number of fish consumption was not illustrated in increasing energy intake (*p*-value = 0.054), protein (*p*-value = 0.083), and fat (*p*-value = 0.151). In Table 7, in the control group, increasing fish consumption did not associate with increasing of energy (*p*-value = 0.051), protein (*p*-value = 0.082), and fat (*p*-value = 0.052).

Table 8 showed that there was a significant improvement in children’s knowledge related to the benefit of fish consumption in the intervention group with 6 session nutrition education for three months and used raised bed pool media (*p*-value < 0.001). However, the control group with only printed educational material also showed a significant increase in knowledge related to the importance of fish consumption (*p*-value = 0.001).

## 4. Discussion

In this study, respondents were specifically learned more about fish consumption, including their benefits, how to prepare them, and how they taste, which may help to facilitate and increase their fish consumption. It is known that schoolchildren need a good diet to develop and grow well. Theory of Planned Behavior (TPB) based intervention using Raised Bed Pool (RBP) in this study revealed that nutrition education within 3-months had given significant effect on students’ behavior which consist of attitude, subjective norm, perceived behavioral control, knowledge, and the amount towards fish consumption. This approach was proven in the previous study that after intensive 3-months nutrition education through the workshop, utilization video, and photovoice has significantly increased fish consumption by more than 5 g per day [21].

Ajzen posited that the change in healthy eating behavior would be easier to be adopted if it is done at the earliest stage of life [22]. Childhood has been identified as a critical period for the development of eating patterns that track to adulthood [23,24,25]. Considering the importance of good nutrition in childhood to achieve healthy growth and development. It will be essential to give children the opportunities to be exposed to healthy food as early as possible [26,27]. A systematic review of related factors influencing children’s eating behavior explained that children’s eating pattern was influenced by the food environment provided by parents and children’s experience [25]. Furthermore, the healthy eating behavior of the children was mirrored by the parent’s eating behavior. One of the healthy eating pattern practices that still needs to be improved in children is fish consumption habits. The latest data from the national survey in 2014 showed that consumption of fish and its product in children was only 70.7 g, lower compared to adult and elderly groups [8], while the availability of fish in Indonesia is quite high. Fish is one of the recommended types of protein consumed by children because of the good fat content that is beneficial for brain development and also high iron content that can reduce the risk of anemia in children.

Our study revealed that the usage of a raised bed pool as a medium for nutrition education intervention successfully improves students’ fish consumption. This highlighted that TPB-nutrition education added by the raised bed pool as an effective nutrition media enhances behavioral change (knowledge, attitude, perceived behavioral control, and intention) related to fish consumption. TPB nutrition education is still necessary as the main effort to convey messages on the importance of consuming fish to school children. Meanwhile, access to RBP adds to hands-on or experiential learning for students. This study added to the growing evidence that school-based nutrition education (NE) programs could lead to moderate increases in fish consumption among children [19,28,29]. Based on the Theory of Planned Behavior construct, the behavior is strongly influenced by intention, which together is influenced by attitudes, subjective norms, and perceived behavioral control [15]. All of the nutrition education activity carried out in this study directed at modifying determinants of behavior (attitudes, subjective norms, perceived behavioral control, and intention of consumption), and consumption of fish. Our study was aligned with a randomized control trial based on TBP construct in 86 children by using 14 sessions of 60 min nutrition education was successfully increase children’s fruit and vegetable intake by first increasing their attitudes, subjective norms, and perceived behavioral control [30]. Furthermore, Kim and Park [29] also explained in their study that garden-based integrated with social cognitive theory elements as one of the behavioral change strategies successfully improved children’s eating behavior for vegetables; which strengthen our results that both TPB and RBP intervention were needed to improve children’s behavior towards fish consumption.

The raised bed pool is adapted from a raised bed garden (RBG). While RBG is the utilization of the garden, RBP is the utilization of the pool. Studies have shown that RBG can significantly increase fresh fruit and vegetable consumption; therefore, RBP is expected to be able to increase fish consumption [28]. In addition, RBP can be a powerful learning medium that is beneficial and worthwhile for both teachers and students. RBP is unique media as it provides potential protein sources from the fish in the pool that is lacking from RBG. Similar to RBG, RBP provides an atmosphere that incorporates hands-on activities and strengthens academic, personal, and social skills. Moreover, it allows children to develop and strengthen their life skills in areas such as nutrition, leadership, and decision making [19].

RBP is a medium that provides authentic and hands-on experience to nature, which is effective at increasing cognitive abilities and higher-order thinking skills. The term “hands-on” gives a more concrete definition of hands-on instruction as well as its influences which young students especially learn through actions, more so than older students, and therefore experience greater benefits from hands-on and action-oriented learning [31]. This helps more students experience more success in their learning because multiple senses reinforce it at once. A meta-analysis study in students in Australia showed that experimental nutritional learning at school was associated with higher effects in healthy consumption in students [30]. It is also supported by the previous study that school garden-enhanced nutrition education could increase students’ willingness to taste vegetables and their vegetable taste ratings [32]. It is further explained that the school garden may increase vegetable intake, but other determinants might take part.

Based on Piaget and other scientists’ theories, they stated that a child’s understanding is developed through his actions on the environment not merely through language [33]. As stated before, experiential education techniques (e.g., raised bed pool) helps children to develop cognitive skills. It will increase children’s’ intrinsic motivation to learn. It can be said that using RBP is an effective learning technique due to their hands-on experience, not to mention with the 3 months. The raised bed pool will be served as a visual reminder to the children that actual learning will magnify the effect of the 6 sessions of behaviorally oriented nutrition education.

One of the strengths of the study is that evidence on the effectiveness of nutrition education intervention using a raised bed pool was scarce relative to the abundance body of knowledge related to the raised bed garden. However, little work has been carried out to address the effectivity of raised bed pool as a nutrition education media and previous work have not comprehensively considered. A limitation in this study is the fact that variation and inequalities between elementary schools whether it was a private or public school. Both elementary schools involved in this study were drawn from public schools located in the low to middle-income population. Generalization of the results of the intervention could be limited due to the current setting of the sample. To limit potential bias, all measurements were performed through a standardized protocol, and all enumerators were trained before data collection. As the study did not measure the actual amount of fish served to the children by their parents, richer parents within the intervention group could induce a bias. Richer parents (intervention group) could afford to buy fish more often, while poorer parents (control group) would be limited. Another limitation was the absence of iron level or anemia status and total energy intake measurement due to budget limitation. Thus, future research might mention this as other dependent variables.

For further studies, another innovative form of nutrition education is needed. Given the complexity of dietary behavior change, a comprehensive study to understand several factors that might affect children’s behavior in consuming fish need to be explored. The education material should be consisting of the number of fishes that should be consumed based on children’s (9–11 years) Recommended Dietary Intake (RDI).

## 5. Conclusions

In conclusion, the 3 months nutrition education intervention based on the theory of planned behavior significantly increase fish consumption among elementary school children. The increased consumption was believed to be related to the increase in children’s knowledge and attitude towards consuming fish. It is suggested to the school setting to continue the nutrition education effort by integrating raised bed pool media with the existing school health program.

## Figures and Tables

**Figure 1 ijerph-17-06970-f001:**
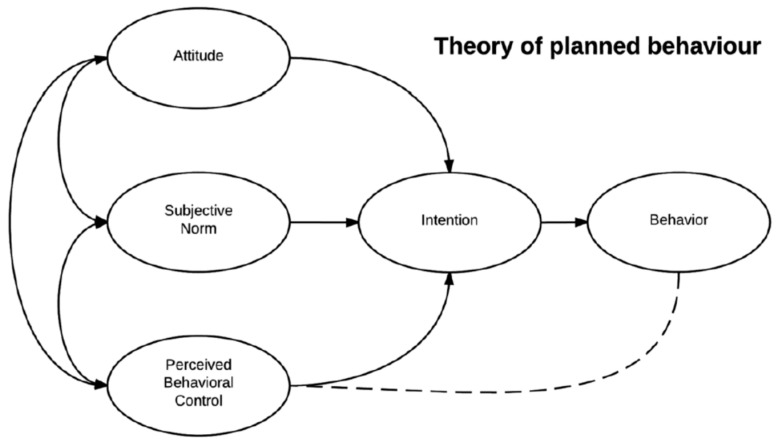
The Construct of Theory of Planned Behavior (adapted from [14] by Robert Orzanna).

**Figure 2 ijerph-17-06970-f002:**
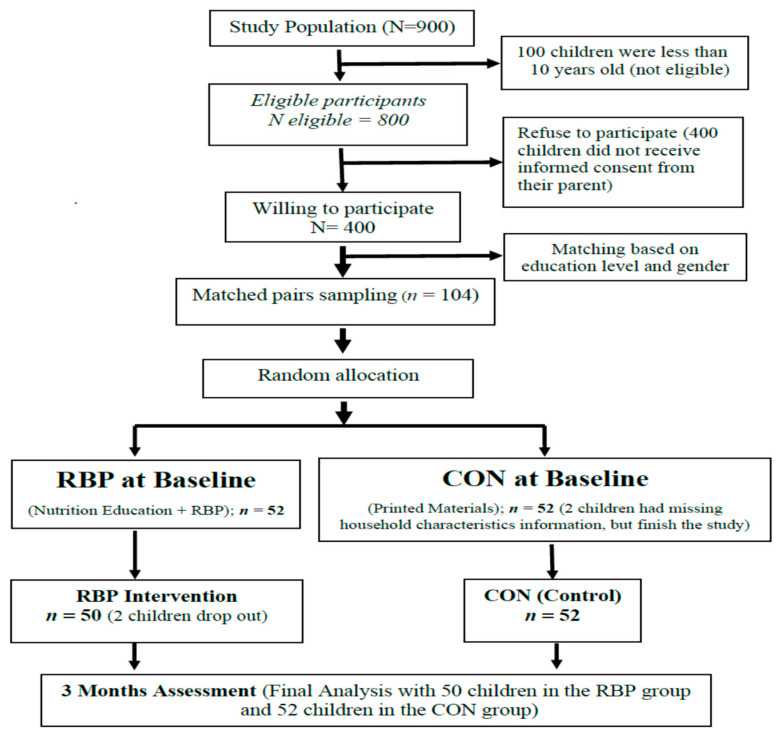
The CONSORT diagram of the study.

**Table 1 ijerph-17-06970-t001:** Children characteristics.

Children Characteristics	Control Group (*n* = 52)		Intervention Group (*n* = 50)		***p-*** **Value**
	Mean	SD	Mean	SD	
Age, years	11.56	0.61	11.76	0.66	0.109
Height, cm	145.04	7.16	144.43	7.07	0.665
Weight, kg	43.76	13.88	39.60	12.07	0.111
BMI, kg/m^2^	20.48	5.14	18.99	4.29	0.118
Total Fat, g	26.52	22.96	23.26	5.99	0.333
Resting Metabolic Rate, Cal/day ^a^	1111.09	309.50	1001.48	260.20	0.056
Fish consumption at Baseline	0.37	0.66	1.30	1.02	<0.001 *

* Statistically significant at alpha = 0.05 based on the independent *t*-test; ^a^ RMR was retrieved through BIA body composition tools.

**Table 2 ijerph-17-06970-t002:** Household characteristics.

Household Characteristics	Control Group (*n* = 50)		Intervention Group (*n* = 50)		*p-*Value *
	*n*	%	*n*	%	
Type of household					0.617
Nuclear family	41	41	39	39	
Extended family	9	9	11	11	
Maternal literacy					0.140
Partially Literate	2	2	6	6	
Literate	48	48	44	44	
Mother’s education					0.212
Did not Finish Elementary	1	1	1	1	
Finish Elementary	11	11	4	4	
Finish Junior High School	2	2	8	8	
Finis Senior High School	27	27	28	28	
Diploma	4	4	3	3	
University	5	5	6	6	
Mother’s occupation					0.437
Housewife	42	42	43	43	
Civil Servant	0	0	1	1	
Company Worker/Employee	1	1	0	0	
Farmers/Labour					
Service Worker	0	0	1	1	
Others	7	7	4	4	
	0	0	1	1	
Household income (IDR)					0.054
<500,000	2	2	5	5	
500,000–1,000,000	13	13	2	2	
>1,000,000–1,500,000	5	5	4	4	
>1,500,000–2,000,000	9	9	16	16	
>2,000,000–2,500,000	3	3	5	5	
>2,500,000–3,000,000	7	7	9	9	
>3,000,000	11	11	9	9	
Household food expenditure					0.243
<500,000	2	2	5	5	
500,000–1,000,000	16	16	2	2	
>1,000,000–1,500,000		5	4	4	
>1,500,000–2,000,000	8	8	16	16	
>2,000,000–2,500,000	3	3	5	5	
>2,500,000–3,000,000	6	6	9	9	
>3,000,000	10	10	9	9	

* The difference in characteristics was analyzed using the Chi-Square test.

**Table 3 ijerph-17-06970-t003:** Change in children’s attitude towards fish consumption after 3 months of intervention.

Attitude Score	Control Group (*n* = 52)			Intervention Group (*n* = 50)		
	Delta Mean	SD	*p*-Value	Delta Mean	SD	*p*-Value
Benefit of fish consumption for health	−0.54	1.91	0.048*	0.26	1.35	0.180
Enjoyment on consuming fish for health	−0.58	2.39	0.087	-0.04	2.09	0.893
Good or bad if we ate fish for our health	−0.25	1.91	0.352	-0.20	2.73	0.607
Benefit of fish consumption for health twice a week	0.19	2.52	0.585	0.40	1.81	0.124
Enjoyment on consuming fish for health twice a week	0.31	2.53	0.385	1.38	2.16	<0.001 *
Good or bad if we ate fish for our health twice a week	0.59	2.38	0.077	0.37	2.79	0.362
Benefit of fish consumption for health everyday	0.46	2.53	0.195	1.22	2.54	0.001 *
Enjoyment on consuming fish for health everyday	1.08	4.84	0.115	0.90	2.05	0.003 *
Good or bad if we ate fish for our health everyday	0.52	2.46	0.134	1.44	2.99	0.001 *

* Statistically significant at alpha = 0.05 based on the paired *t*-test.

**Table 4 ijerph-17-06970-t004:** Change in children’s subjective norm in fish consumption after 3 months of intervention.

Subjective Norm Score	Control Group (*n* = 52)			Intervention Group (*n* = 50)		
	Delta Mean	SD	*p*-Value	Delta Mean	SD	*p*-Value
Parental advice to eat fish	−0.23	1.42	0.248	0.16	1.66	0.498
Peers/friends advice to eat fish	−0.27	1.74	0.269	0.44	1.87	0.138
Teacher advice to eat fish	−0.31	1.59	0.169	−0.14	1.47	0.504
Doctor advice to eat fish	−0.33	1.59	0.145	−0.22	1.46	0.292
Parental consent to eat more fish	−0.35	1.44	0.089	−0.10	1.50	0.640
Peers/friends consent to eat more fish	0.31	2.21	0.320	0.32	1.65	0.176
Teacher consent to eat more fish	−0.12	1.35	0.541	0.06	1.47	0.775
Doctor consent to eat more fish	−0.17	1.74	0.475	−0.14	1.54	0.523

Statistically significant at alpha = 0.05 based on the paired *t*-test.

**Table 5 ijerph-17-06970-t005:** Change in children’s perceived behavioral control towards fish consumption after 3 months of intervention.

Perceived Behavioural Control Score	Control Group (*n* = 52)			Intervention Group (*n* = 50)		
	Delta Mean	SD	*p*-Value	Delta Mean	SD	*p*-Value
Perception that increasing fish consumption in one month is easy	−0.12	1.75	0.636	−0.36	2.14	0.239
I can increase my fish consumption if I wanted to	−0.02	2.65	0.958	0.32	1.96	0.255
I can increase my fish consumption if I know the benefit for it	−0.52	1.85	0.049 *	0.18	1.54	0.411

* Statistically significant at alpha = 0.05 based on the paired *t*-test.

**Table 6 ijerph-17-06970-t006:** Change in children’s intention towards fish consumption after 3 months of intervention.

Intention Score	Control Group (*n* = 52)			Intervention Group (*n* = 50)		
	Delta Mean	SD	*p*-Value	Delta Mean	SD	*p*-Value
I intent to eat fish	−0.15	1.76	0.532	0.02	1.57	0.929
I will try to eat fish	−0.25	2.02	0.376	−0.30	1.74	0.229
I am planning to eat fish	−0.54	1.93	0.049 *	−0.28	1.85	0.290

* Statistically significant at alpha = 0.05 based on the paired *t*-test.

**Table 7 ijerph-17-06970-t007:** Change in children’s fish consumption after 3 months intervention.

Variable	Control Group (*n* = 52)			Intervention Group (*n* = 50)		
	Delta Mean	SD	*p*-Value	Delta Mean	SD	*p*-Value
Number of fishes consumed	0.19	1.03	0.184	0.58	1.74	0.022 *
Energy intake from fish consumed (kcal/day)	11.73	42.42	0.051	21.87	78.33	0.054
Protein intake from fish consumed in a day (gr/day)	1.95	7.90	0.082	3.38	13.49	0.083
Fat intake from fish consumed in a day (gr/day)	0.38	1.37	0.052	0.84	4.07	0.151

* Statistically significant at alpha = 0.05 based on the paired *t*-test.

**Table 8 ijerph-17-06970-t008:** Change in children’s knowledge of the importance of fish consumption after 3 months of intervention.

Variable	Control Group (*n* = 52)			Intervention Group (*n* = 50)		
	Delta Mean	SD	*p*-Value	Delta Mean	SD	*p*-Value
Knowledge Score	1.29	2.65	0.001 *	1.70	2.54	<0.001 *

* Statistically significant at alpha = 0.05 based on the paired *t*-test.

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
