# Peer review of "Nutrition Education Intervention Increases Fish Consumption among School Children in Indonesia: Results from Behavioral Based Randomized Control Trial"

_ijerph, 2020, doi:10.3390/ijerph17196970_

Round 1

Reviewer 1 Report

The authors of manuscript, IJERPH 924295, Nutrition Education Intervention Increase[s?] Fish Consumption among School Children in Indonesia: Results from Behavioral Based Randomized Controlled Trial, address the important problem of iron deficiencies in children living in a large metropolitan area, Surabaya, Indonesia. Fish consumption is targeted for its high rate of iron absorption. A nutrition education intervention based on the Theory of Planned Behavior (TPB), coupled with a Raised Bed Pool (RBP) project in school, are tested in combination as a potential means of increasing fish consumption.

Introduction:

  • Line 46: The acronym “IFLS” is not introduced.
  • Lines 56-58: The “times/week” descriptors in parentheses should follow the time-referenced words, i.e., “occasionally (2-3 times/week)” and “frequently (4-7 times/week).”
  • TPB addresses the potential for changes in participants’ “desire,” “intention,” “attitudes,” “perceived behavioral control,” and “skills” for achieving the sought-after behavioral outcome of increased fish consumption.
  • I believe line 95 should begin, “Analogous to involving children…”
  • Lines 122-124: This sentence is unclear, given the use of the word “overcome.” It reads as if the intervention is designed to “overcome fish intervention and anemia outcomes in school children.” I’m not sure what that means, but I believe the intervention is designed to improve fish consumption, which should, in turn, reduce anemia in school children.

Materials and Methods:

  • The study was conducted in a low to medium income school district where protein (including fish) availability and consumption tends to be low (if I read this correctly).
  • It sounds like 500 of 900 children were deemed ineligible prior to further screening. What accounted for so many children being ineligible? This should be explained in this section.
    • Regarding inclusion/exclusion criteria: What specific types of “special diets” would preclude participation?
    • Does this include weight-loss diets?
    • Were medical or behavioral diagnoses or characteristics used to exclude children? For example, metabolic syndrome, obesity, attention deficit disorder, etc.
  • Lines 135-136. The term “dropout” suggests (to me) that children or their mother could actively discontinue participation. However, it sounds like participants were withdrawn (by study investigators) as participants if they missed more than 50% (more than 3 of the 6) of the sessions. Is that true? If so, is “withdrawn” a better term?
  • Details on the procedure for contacting families and consenting or assenting children and mothers, is missing. How was this accomplished? Who on the study team implemented screening and enrollment (including consenting)? Details can be brief, but in my judgment should be provided.
  • Independent variables: Several are listed (lines 161-163).
    • It would be clearer if they were organized as moderating (SES), mediating and outcome variables. For example, parent education, weight status, etc., would be moderating variables. Intentions, perceived behavioral control, etc., would be mediating variables (cognitive mediators). And fish consumption would be considered as the behavioral outcome.
    • The nutrition education intervention is an independent variable, but it is not listed as such. The control condition is an independent variable as well, because children and mothers were given printed materials. The nutrition education intervention would be expected to influence the key outcome (dependent variable) of fish consumption by altering mediating variables (from the TPB), resulting in actual behavior change in the form of consumption of fish. Nutrition education would hypothetically result more reported fish consumption than the use of printed materials alone.
    • Stated hypotheses are needed at the end of the Introduction section.
    • Weight and body composition were assessed, presumably as moderating variables? There is no rationale for the inclusion of weight and body composition as a measure, and obesity wasn’t included in the inclusion/exclusion criteria.
    • More information is needed on the assessment of child and parent characteristics. What are some sample questions in the parent questionnaire? The reader would benefit. Please also provide a citation for the child instrument used, and any citation that applies to the basis of the parent instrument.
  • Dependent variable: “Fish consumption” appears to be the single dependent variable.
    • Iron count or anemia status was not measured in this study. This might be mentioned as a limitation or a call for future research in the Discussion section.
    • Fish consumption was measured using three-day food diaries completed by the children. What specific form (with citation) was used with these 10-12 year old children? Did the recording (as is done typically) require the child to record all foods consumed on the three days? Did the children’s mother, or teachers, assist with recording? What processing software or materials (with citation) was used to analyzed the three-day records? Were all foods analyzed, or just fish? Who on the research team analyzed the diaries? What was their training (lines 166-167 are unclear)?
    • Listing examples of the questions used in the TPB surveys would be helpful.
  • RBP participants were given monetary incentives. Was that given to the CON participants as well?
  • Was there any further contact between the study staff and the CON participants following the delivery of printed materials?

Results:

  • Lines 270-271: The statement, “…significantly changed behavior was found in the intervention group,” is misleading (not intentionally, I’m sure!). The measures in Tables 3-6 are comprised of attitudes, subject norms, perceived behavioral control, and intentions. These are important mediators of behavior change, but they are not measures of the actual behavior. The actual behavior that needs to change is fish consumption (Table 7), which is measured through self-report in the three-day diaries. The TPB measures are mediators of the study outcome of behavior change.
  • This is all stated more accurately in the Conclusion section: “The increased consumption was believed to be related to the increase in the children’s knowledge and attitude towards consuming fish.” The wording of this sentence more accurately describes the relationship between mediators of action (knowledge and attitude) and the actual behavior change/action of consuming fish.
  • More comments on results appear below.

Discussion:

  • See also lines 325-329 for reference to TPB mediators as behavior.
  • It’s interesting that the multiple TPB variables reported in Tables 4, 5, and 6 (and in Table 3, with the exception of enjoyment of consuming fish) did not change significantly as a result of a TBP-based (combined with raised bed pool) intervention. Given that the suspected importance of a TBP-based approach, as stated in the Introduction, and the resources required to deliver it as part of an intervention, it would seem important that the authors comment on the need for a TPB-basis for the intervention at all? Would the raised bed pool activity, as a means to introduce children to fish, be sufficient for children to enjoy eating more fish? This seems like a very important item for discussion.
  • Lines 346-347: If I read the manuscript correctly, the intervention combined two variables - access to the raised bed pool, and TPB-based nutrition education sessions. The full intervention is called RBP. Access to the raised bed pool alone was not assessed in isolation (without the nutrition education components). I believe this should be made clearer in the discussion comments and throughout. You might just make it clearer that the term “RBP” as the label of the active intervention in your study includes both physical access to a pool itself and the TBP-based nutrition education curriculum/lessons. The raised bed pool educational medium sounds fascinating, and it combines well with hands-on education, as the authors correctly assert. But the nature of the hands-on education can vary; in this case it was on nutrition education for fish consumption, as opposed to other learning goals, include farming practices, etc. Again, in your view as authors, are the TPB-based components necessary?

Author Response

Response to Reviewers

Title

:

Nutrition Education Intervention Increase Fish Consumption among School Children in Indonesia: Results from Behavioral Based Randomized Control Trial

Authors

:

Trias Mahmudiono, Triska Susila Nindya, Qonita Rachmah, Calista Segalita, Luh Ade Wiradnyani

No

Section

Reviewer Corrections

Page

Author’s Revision

Reviewer 1

General Comment

The authors of manuscript, IJERPH 924295, Nutrition Education Intervention Increase[s?] Fish Consumption among School Children in Indonesia: Results from Behavioral Based Randomized Controlled Trial, address the important problem of iron deficiencies in children living in a large metropolitan area, Surabaya, Indonesia. Fish consumption is targeted for its high rate of iron absorption. A nutrition education intervention based on the Theory of Planned Behavior (TPB), coupled with a Raised Bed Pool (RBP) project in school, are tested in combination as a potential means of increasing fish consumption.

Thank you for your valuable inputs and feedbacks.

1

Title

2

Abstract

3

Introduction

1.       Line 46: The acronym “IFLS” is not introduced.

2.       Lines 56-58: The “times/week” descriptors in parentheses should follow the time-referenced words, i.e., “occasionally (2-3 times/week)” and “frequently (4-7 times/week).”

3.       TPB addresses the potential for changes in participants’ “desire,” “intention,” “attitudes,” “perceived behavioral control,” and “skills” for achieving the sought-after behavioral outcome of increased fish consumption.

4.       I believe line 95 should begin, “Analogous to involving children…”

5.       Lines 122-124: This sentence is unclear, given the use of the word “overcome.” It reads as if the intervention is designed to “overcome fish intervention and anemia outcomes in school children.” I’m not sure what that means, but I believe the intervention is designed to improve fish consumption, which should, in turn, reduce anemia in school children.

1.       IFLS acronym was stated (line 44)

2.       Revised as suggested (line 54-55)

3.       Revised as suggested (line 77-79)

4.       Revised as suggested (line 93)

5.       Agreed; we changed the sentence to ‘Based on the description above, this research would like to evaluate the effectiveness of RBP as a media of nutrition education intervention to increase fish consumption as part of anemia prevention efforts in school children

4

Materials and Methods

1.       The study was conducted in a low to medium income school district where protein (including fish) availability and consumption tends to be low (if I read this correctly).

2.       It sounds like 500 of 900 children were deemed ineligible prior to further screening. What accounted for so many children being ineligible? This should be explained in this section.

3.       Regarding inclusion/exclusion criteria: What specific types of “special diets” would preclude participation?

Does this include weight-loss diets?

4.       Were medical or behavioral diagnoses or characteristics used to exclude children? For example, metabolic syndrome, obesity, attention deficit disorder, etc.

5.       Lines 135-136. The term “dropout” suggests (to me) that children or their mother could actively discontinue participation. However, it sounds like participants were withdrawn (by study investigators) as participants if they missed more than 50% (more than 3 of the 6) of the sessions. Is that true? If so, is “withdrawn” a better term?

6.       Details on the procedure for contacting families and consenting or assenting children and mothers, is missing. How was this accomplished? Who on the study team implemented screening and enrollment (including consenting)? Details can be brief, but in my judgment should be provided.

7.       Independent variables: Several are listed (lines 161-163).It would be clearer if they were organized as moderating (SES), mediating and outcome variables. For example, parent education, weight status, etc., would be moderating variables. Intentions, perceived behavioral control, etc., would be mediating variables (cognitive mediators). And fish consumption would be considered as the behavioral outcome.

8.       The nutrition education intervention is an independent variable, but it is not listed as such. The control condition is an independent variable as well, because children and mothers were given printed materials. The nutrition education intervention would be expected to influence the key outcome (dependent variable) of fish consumption by altering mediating variables (from the TPB), resulting in actual behavior change in the form of consumption of fish. Nutrition education would hypothetically result more reported fish consumption than the use of printed materials alone.

9.       Stated hypotheses are needed at the end of the Introduction section.

10.    Weight and body composition were assessed, presumably as moderating variables? There is no rationale for the inclusion of weight and body composition as a measure, and obesity wasn’t included in the inclusion/exclusion criteria.

11.    More information is needed on the assessment of child and parent characteristics. What are some sample questions in the parent questionnaire? The reader would benefit. Please also provide a citation for the child instrument used, and any citation that applies to the basis of the parent instrument.

12.    Dependent variable: “Fish consumption” appears to be the single dependent variable.

13.    Iron count or anemia status was not measured in this study. This might be mentioned as a limitation or a call for future research in the Discussion section.

14.    Fish consumption was measured using three-day food diaries completed by the children. What specific form (with citation) was used with these 10-12 year old children? Did the recording (as is done typically) require the child to record all foods consumed on the three days? Did the children’s mother, or teachers, assist with recording? What processing software or materials (with citation) was used to analyzed the three-day records? Were all foods analyzed, or just fish? Who on the research team analyzed the diaries? What was their training (lines 166-167 are unclear)?

15.    Listing examples of the questions used in the TPB surveys would be helpful.

16.    RBP participants were given monetary incentives. Was that given to the CON participants as well?

17.    Was there any further contact between the study staff and the CON participants following the delivery of printed materials?

1.       Correct, we added that information on the sentence (line 137)

2.       Thank you for your concern, it was a typo/mistake on our end. Out of 900, 800 were eligible for the study.

3.       We put an example of special diet (line 133-134)

4.       We did not specifically diagnose medical or behavioral diagnoses to exclude the children; but we did put in the criteria that children with special diet which usually goes by specific medical condition did not included in our study.

5.       We changed to “withdrawn” instead of dropout (line 134)

6.       We added the information (line 240-244)

7.       Revised as suggested (line 158-161)

8.       Thank you for your suggestion, we added the independent ariables in the method section (line 157-161)

9.       Added as suggested (line 128-129)

10.    We added an information the reason of weight and body composition data collected (line 176)

11.    Information was added (line181-184)

Citation was added it was based on this study “16.             Hutchinson, J.; Christian, M. S.; Evans, C. E. L.; Nykjaer, C.; Hancock, N.; Cade, J. E. Evaluation of the Impact of School Gardening Interventions on Children’s Knowledge of and Attitudes towards Fruit and Vegetables. A Cluster Randomised Controlled Trial. Appetite 2015, 91, 405–414. https://doi.org/10.1016/j.appet.2015.04.07

12.    Agreed

13.    Added as limitation (line 423-425)

14.    We clarify this section as suggested (line 163-169).

We add citation “Crawford, P. B., Obarzanek, E., Morrison, J., & Sabry, Z. I. (1994). Comparative advantage of 3-day food records over 24-hour recall and 5-day food frequency validated by observation of 9-and 10-year-old girls. Journal of the American Dietetic Association, 94(6), 626-630.

15.    We list some questions used in TPB surveys (line 199-204)

16.    Yes. We added the information regarding this in the manuscript (line 226)

17.    No. We added the information regarding this in the manuscript (line 220-221)

5

Results

Lines 270-271: The statement, “…significantly changed behavior was found in the intervention group,” is misleading (not intentionally, I’m sure!). The measures in Tables 3-6 are comprised of attitudes, subject norms, perceived behavioral control, and intentions. These are important mediators of behavior change, but they are not measures of the actual behavior. The actual behavior that needs to change is fish consumption (Table 7), which is measured through self-report in the three-day diaries. The TPB measures are mediators of the study outcome of behavior change.

This is all stated more accurately in the Conclusion section: “The increased consumption was believed to be related to the increase in the children’s knowledge and attitude towards consuming fish.” The wording of this sentence more accurately describes the relationship between mediators of action (knowledge and attitude) and the actual behavior change/action of consuming fish.

More comments on results appear below.

Agree, We Change in children’s attitude towards fish consumption after 3 months intervention

Revised as suggested (line 280)

6

Discussion

1.       See also lines 325-329 for reference to TPB mediators as behavior.

2.       It’s interesting that the multiple TPB variables reported in Tables 4, 5, and 6 (and in Table 3, with the exception of enjoyment of consuming fish) did not change significantly as a result of a TBP-based (combined with raised bed pool) intervention. Given that the suspected importance of a TBP-based approach, as stated in the Introduction, and the resources required to deliver it as part of an intervention, it would seem important that the authors comment on the need for a TPB-basis for the intervention at all? Would the raised bed pool activity, as a means to introduce children to fish, be sufficient for children to enjoy eating more fish? This seems like a very important item for discussion.

3.       Lines 346-347: If I read the manuscript correctly, the intervention combined two variables - access to the raised bed pool, and TPB-based nutrition education sessions. The full intervention is called RBP. Access to the raised bed pool alone was not assessed in isolation (without the nutrition education components). I believe this should be made clearer in the discussion comments and throughout. You might just make it clearer that the term “RBP” as the label of the active intervention in your study includes both physical access to a pool itself and the TBP-based nutrition education curriculum/lessons. The raised bed pool educational medium sounds fascinating, and it combines well with hands-on education, as the authors correctly assert. But the nature of the hands-on education can vary; in this case it was on nutrition education for fish consumption, as opposed to other learning goals, include farming practices, etc. Again, in your view as authors, are the TPB-based components necessary?

o   Utk poin 2, aku notes di body of email ya

o   Utk poin 3, setuju untuk menggunakan istilah yang konsisten. Full intervention dinamakan RBP yang isinya sebetulnya RBP+nutrition education sessions.

o   Untuk pertanyaan reviewer apakah TPB-nutrition education itu perlu (karena kita tidak punya kelp intervensi yang isinya RBP saja), memang agak tricky ya, karena tidak ada kegiatan yang beririsan antara kelp intervensi dan kontrol. Kelp intervensi menerima exposure akses dan ikut menjaga kolam serta nutrition education yang lebih engaging, sedangkan kelp kontrol menerima kegiatan yg pasif (sets of printed educational materials, including info-graphic, comic, and recipe book.

Menurutku, si TPB-nutrition education itu tetep perlu sebagai upaya utama untuk menyampaikan pesan-pesan pentingnya mengkonsumsi ikan pada siswa. Sedangkan akses ke RBP menambah experiential learning ke siswa sehingga mereka lebih tertarik. Untuk memberi ide menjelaskan hal ini bisa merujuk ke references ttg peran2 sejenis dalam behavior changes, misalnya intervensi nutrition education yang ditambah dengan akses ke kebun gizi di sekolah) >> Added in line 357-362 (mohon dikoreksi jika salah) and line 371 – 375

Tambahan sitasi: Kim, S. O., & Park, S. (2020). Garden-Based Integrated Intervention for Improving Children’s Eating Behavior for Vegetables. International Journal of Environmental Research and Public Health, 17(4), 1257.

Reviewer 2 Report

Thank you to the authors for this manuscript. The study methods and results descriptions are well done, and the manuscript is an interesting public health approach to hidden hunger. My minor comments are below:

I would suggest that an English editor review this manuscript

Line 28: what does improve their internalization mean?

Line 29: might be able to remove ‘planned behavior aspects’

Line 32-33: sentence clarification needed

Lines 61-69: might include the heme/iron content of fish on average here

Do the authors have permission for figure 1, or is it self created?

Could authors spend a little time clarifying ‘love eating fish’ vs. RBP? Are they linked somehow?

Line 116: citation?

Lines 126-130: can this be moved into methodology?

Line 135: what does special diet mean? That they would not consume fish?

Regarding inclusion/exclusion- how did authors get from 400 to 104? Can they include why the 296 students were excluded from the research?

In table 1, how was the RMR calculated?

To clarify- was there a significant increase in enjoyment of the control compared to the intervention group, or am I misreading the data?

Author Response

No

Section

Reviewer Corrections

Page

Author’s Revision

Reviewer 2

General Comment

Thank you to the authors for this manuscript. The study methods and results descriptions are well done, and the manuscript is an interesting public health approach to hidden hunger. My minor comments are below:

I would suggest that an English editor review this manuscript

Line 28: what does improve their internalization mean?

Line 29: might be able to remove ‘planned behavior aspects’

Line 32-33: sentence clarification needed

Lines 61-69: might include the heme/iron content of fish on average here

Do the authors have permission for figure 1, or is it self created?

Could authors spend a little time clarifying ‘love eating fish’ vs. RBP? Are they linked somehow?

Line 116: citation?

Lines 126-130: can this be moved into methodology?

Line 135: what does special diet mean? That they would not consume fish?

Regarding inclusion/exclusion- how did authors get from 400 to 104? Can they include why the 296 students were excluded from the research?

In table 1, how was the RMR calculated?

To clarify- was there a significant increase in enjoyment of the control compared to the intervention group, or am I misreading the data?

Line 28: internalization means make (attitudes or behavior) part of one's nature by learning or unconscious assimilation.

Line 29: Deleted as suggested

Line 32-33: revised as suggested

Line 61: iron content in food was added

Figure 1 was originated from BAndura’s guide citation no #14

We deleted word ”love”, just to make clear that RBP was developed to inrease childrens’ intention to eat fish

Line 116 is author’s assumption, there was no citation

Line 126 : moved as suggested (line 127-131)

Line 135: we added more information regarding special diets (line 133)

RMR was retrieved from BIA body composition tools (information was added below the table)

Reviewer 3 Report

The study submitted by Mahmudiono et alia describes how nutritional interventions using out-of-the-ground fish pools could be beneficial to increase knowledge of fish, intention to eat fish and consumption of fish in children. Using a RCT protocol, authors find a successful increase of fish consumption following intervention. However, a significant bias could reside in the fact authors do not show baseline (pre-intervention) eating habits (fish intake, serving, ration, etc…), to be able to compare with final intervention. This is a serious concern and is detailed below (point 3 in the major remarks paragraph), together with further remarks. Moreover, ethical concerns arose regarding animal welfare (detailed below).

Major remarks.

1/ Figure 1 is identical to a figure from Wikipedia. I am unsure whether this image is granted for free/unlimited reuse. Please avoid using this figure, as this can be perceived as plagiarism. Alternatively, if authors were granted to re-use such a figure, please give the original reference.

2/ The “ethics” paragraph should be placed first within the methods section. Besides, since this RCT used catfish, authors should describe the following points :
a/ How were the fish acquired ? (farmed ? from the wild ?).
b/ How many fish were included in each pool ?
c/ How many pools were present in total ?
d/ What were the housing conditions for the fish ?
e/ Since animals are used, have authors obtained ethical approval for the use of fish in the RTC ?
f/ Finally, what was done with the fish after the end of the trial ? (euthanasia ? re-introduced into the wild ? consumption ?).

3/ The authors suppose that the intervention with raised bed pool was indeed successful in increasing fish consumption in children, as seen in Table 8. However, a heavy limitation could reside in increasing amount of fish served to the children by their parents, unrelated to the intervention performed in the study. Are authors implying that children from the intervention group are asking their parents to eat/buy more fish as a result of the interventions ? Have authors checked for such a possibility ? Did the authors check (with questionnaires) baseline fish serving in both groups, compared to servings following the intervention ? Moreover, seeing the household income distribution presented in Table 2, richer parents are found within the intervention group, which almost reaching statistical significance (p=0.054). This could induce a bias, since richer parents (intervention group) could afford buying fish more often, while poorer parents (control group) would be limited.

4/ The English grammar needs to be thoroughly revised throughout.

5/ Figure 2, Table 1 and Tables 3-8 all present the final groups with n=52 (control) and n=50 (intervention) participants. However, Table 2 only present household characteristics in n=50 (control) and n=52 (interventions). Authors should correct or mention the reason(s) for such a discrepancy.

Minor remarks:

1/ Introduction, page 3, lines 117-118 “it is assumed that RBP have a higher appeal to actively involved the school children than the raised bed garden”. Lack of foundation. Please include reference(s).

2/ Introduction, page 4, lines 128-129 “According to Surabaya Regional Statistical Survey, Sidotopo had higher migration rate compare to other subdistricts”. Please add references.

3/ Methods, page 4, lines 138-140 “Generally, in the region with the economic status of the population low to medium income, the possibility of animal protein consumption was low, including fish consumption.”. Please add references.

4/ Methods, page 4, lines 151-152 ”Then, taking into account the design effect 1+ (p (m + 1) using cluster size 30”. Please explain the equation given herein, especially the variables p and m.

5/ Results. Table 1 should also include BMI values for each group. This can be easily performed, since authors already collected weight and height of all participants.

6/ Results. Authors should explain more precisely why money was given to the participants. Besides, it is not clear if such a protocol was performed only once at the beginning of the study or if it was more frequent. Please amend accordingly.

7/ Results. Table 7 should be amended. What are the units for energy intake?. Please also include total energy intake, regardless of food source, to control for more abundant meal portions in one group or another.

8/ Results. It would be useful for readers to compare household incomes with the average monthly salary in Indonesia. Could authors provide such an information ?

9/ Discussion, page 11, lines 343-345, please include references. Please also briefly outline the importance of dietary polyunsaturated fatty acids (PUFAs) found in fatty fish, as well as a few RCT where interventions with n-3 PUFAs was performed.

Author Response

No

Section

Reviewer Corrections

Page

Author’s Revision

Reviewer 3

General Comment

The study submitted by Mahmudiono et alia describes how nutritional interventions using out-of-the-ground fish pools could be beneficial to increase knowledge of fish, intention to eat fish and consumption of fish in children. Using a RCT protocol, authors find a successful increase of fish consumption following intervention. However, a significant bias could reside in the fact authors do not show baseline (pre-intervention) eating habits (fish intake, serving, ration, etc…), to be able to compare with final intervention. This is a serious concern and is detailed below (point 3 in the major remarks paragraph), together with further remarks. Moreover, ethical concerns arose regarding animal welfare (detailed below).

1/ Figure 1 is identical to a figure from Wikipedia. I am unsure whether this image is granted for free/unlimited reuse. Please avoid using this figure, as this can be perceived as plagiarism. Alternatively, if authors were granted to re-use such a figure, please give the original reference.

2/ The “ethics” paragraph should be placed first within the methods section. Besides, since this RCT used catfish, authors should describe the following points :

a/ How were the fish acquired ? (farmed ? from the wild ?).

b/ How many fish were included in each pool ?

c/ How many pools were present in total ?

d/ What were the housing conditions for the fish ?

e/ Since animals are used, have authors obtained ethical approval for the use of fish in the RTC ?

f/ Finally, what was done with the fish after the end of the trial ? (euthanasia ? re-introduced into the wild ? consumption ?).

3/ The authors suppose that the intervention with raised bed pool was indeed successful in increasing fish consumption in children, as seen in Table 8. However, a heavy limitation could reside in increasing amount of fish served to the children by their parents, unrelated to the intervention performed in the study. Are authors implying that children from the intervention group are asking their parents to eat/buy more fish as a result of the interventions ? Have authors checked for such a possibility ? Did the authors check (with questionnaires) baseline fish serving in both groups, compared to servings following the intervention ? Moreover, seeing the household income distribution presented in Table 2, richer parents are found within the intervention group, which almost reaching statistical significance (p=0.054). This could induce a bias, since richer parents (intervention group) could afford buying fish more often, while poorer parents (control group) would be limited.

4/ The English grammar needs to be thoroughly revised throughout.

5/ Figure 2, Table 1 and Tables 3-8 all present the final groups with n=52 (control) and n=50 (intervention) participants. However, Table 2 only present household characteristics in n=50 (control) and n=52 (interventions). Authors should correct or mention the reason(s) for such a discrepancy.

1.        We add ref #14 in the figure to avoid issue of plagiarism

2.        Ethic section was moved to paragraph 2 method section

2a-2d. added in line 238

2e yes the ethical approval from Universitas Airlangga inlude concern related to animal wellfare.

2f. After the end of the study the fish was continued to be raised by schools guard/personnel. Few months back we actually got picture from the school that they expanding the raised bed pool to incorporate with hidrophonic farming.

3.        We did not check or making any observation at the household level. We agree that this was part of serious limitation in our study. We added this limitation in the end of discussion.

4.        Thank you for your suggestion. We revised some gramatical error.

5.        Revised as suggested

1

Title

2

Abstract

3

Introduction

1/ Introduction, page 3, lines 117-118 “it is assumed that RBP have a higher appeal to actively involved the school children than the raised bed garden”. Lack of foundation. Please include reference(s).

2/ Introduction, page 4, lines 128-129 “According to Surabaya Regional Statistical Survey, Sidotopo had higher migration rate compare to other subdistricts”. Please add references.

Agree

2. reference added but paragraph was put in method as suggested by other reviewers (paragraph 1).

4

Material and methods

3/ Methods, page 4, lines 138-140 “Generally, in the region with the economic status of the population low to medium income, the possibility of animal protein consumption was low, including fish consumption.”. Please add references.

4/ Methods, page 4, lines 151-152 ”Then, taking into account the design effect 1+ (p (m + 1) using cluster size 30”. Please explain the equation given herein, especially the variables p and m.

3.       Thank you for your suggestion. Refference added.

4.       Explanation was added m is the number of observations in each cluster and ρ (rho) is the intra-cluster correlation

5

Results

5/ Results. Table 1 should also include BMI values for each group. This can be easily performed, since authors already collected weight and height of all participants.

6/ Results. Authors should explain more precisely why money was given to the participants. Besides, it is not clear if such a protocol was performed only once at the beginning of the study or if it was more frequent. Please amend accordingly.

7/ Results. Table 7 should be amended. What are the units for energy intake?. Please also include total energy intake, regardless of food source, to control for more abundant meal portions in one group or another.

8/ Results. It would be useful for readers to compare household incomes with the average monthly salary in Indonesia. Could authors provide such an information ?

5.       Agree

6.       We did not given money to the participants as it written ”stationeries worth USD $1 or about IDR 15,000”, so we gave stationaries as an incentive. However, since it could be bias for readers, we might need to delete the money worth.

7.       Revised as suggested

8.       Revised as suggested (line 277)

6

Discussion

9/ Discussion, page 11, lines 343-345, please include references. Please also briefly outline the importance of dietary polyunsaturated fatty acids (PUFAs) found in fatty fish, as well as a few RCT where interventions with n-3 PUFAs was performed.

9.       Agree. We added citation. Kim, S. O., & Park, S. (2020). Garden-Based Integrated Intervention for Improving Children’s Eating Behavior for Vegetables. International Journal of Environmental Research and Public Health, 17(4), 1257.

7

Conclusion

8

Acknowledg-

ments

9

References

10

Etc

Round 2

Reviewer 1 Report

The authors have been responsive to the review in their revision, and the manuscript is much improved. Some editing for English language is needed, and more information is needed on the path from a population size of 900 to the end sample sizes in each group. Here are a few final comments on the revised manuscript:

1) I still don't understand the sample size determinants. The study population was 900 children. The authors state that out of 900, 800 were deemed eligible. First, it seems odd that the number of eligible participants was exactly 800. But also, Figure 2 lists 400 (not 800) as eligible. In my opinion, Figure 2 must be presented as a full CONSORT diagram in which the reductions in participant numbers, and the reasons for each reduction down to the numbers in each study arm (group), are presented.

2) Lines 159-160 should reflect that the raised bed pool (RBP) was part of the nutrition education intervention. One possibility is the following edit, "The independent variables in the study were the nutrition education intervention (utilizing the RBP) and..."

3) Lines 200-201: I guess I still need to better understand the rationale for including body weight (BW) and body composition. The authors added, "...to understand the distribution of school children's nutritional status." Certainly, over-nutrition and under-nutrition affect BW and body composition, but that happens in complex and individualized ways. I like the focus on the effects of fish consumption on anemia because it's a direct pathway. I'm just not sure how significantly BW and body composition measures would be affected by fish consumption alone.

Author Response

Response to Reviewer 1

The authors have been responsive to the review in their revision, and the manuscript is much improved. Some editing for English language is needed, and more information is needed on the path from a population size of 900 to the end sample sizes in each group. Here are a few final comments on the revised manuscript:

1) I still don't understand the sample size determinants. The study population was 900 children. The authors state that out of 900, 800 were deemed eligible. First, it seems odd that the number of eligible participants was exactly 800. But also, Figure 2 lists 400 (not 800) as eligible. In my opinion, Figure 2 must be presented as a full CONSORT diagram in which the reductions in participant numbers, and the reasons for each reduction down to the numbers in each study arm (group), are presented.

Response:

Thank you for your suggestion. We revised the CONSORT in to a full CONSORT diagram with detailed reduction in participants and its reasons.

2) Lines 159-160 should reflect that the raised bed pool (RBP) was part of the nutrition education intervention. One possibility is the following edit, "The independent variables in the study were the nutrition education intervention (utilizing the RBP) and..."

Response:

Thank you for your suggestion. We revised the sentence in line 159-160 as per your suggestion.

3) Lines 200-201: I guess I still need to better understand the rationale for including body weight (BW) and body composition. The authors added, "...to understand the distribution of school children's nutritional status." Certainly, over-nutrition and under-nutrition affect BW and body composition, but that happens in complex and individualized ways. I like the focus on the effects of fish consumption on anemia because it's a direct pathway. I'm just not sure how significantly BW and body composition measures would be affected by fish consumption alone.

Response:

Thank you for your suggestion. We additional explanation to revised the sentence in line 200-201. The revised version can be found in line 200-204.

Reviewer 3 Report

Dear Authors,

Thank you for revising your manuscript. I have carefully read all of your answers, which, for most, are satisfactory.

A few more alterations are needed before publication.

First, the English grammar should be corrected throughout. The editors at IJERPH will hopefully help you in doing so during the proofing of your manuscript.

Second, regarding my comments on animal welfare, you cited the Declaration of Helsinki, which concerns for 99% Human welfare. Although animal welfare is, indeed, mentioned in the Helskinky Declaration, it is not sufficient to mention such a protocol since animal welfare is not the purpose of the Helsinki declaration.
Moreover, you have stated that information regarding points 2a-2e are included in line 238, which I cannot find. Moreover, point 2f is still not answered. Please include these details in the manuscript.

Third, regarding Figure 1, you have now added reference number 14, which is the original paper by Ajzen and Fishbein on planned behaviour. However, your Figure 1 is an adaptation from Figure 5.3 of Ajzen and Fishbein made by Robert Orzanna, under a CC-BY-SA-4.0 licence, which grants free permission to re-use, provided that you cite the authors. Therefore, the caption should read something like "adapted from [14] by Robert Orzanna". Please revise.

Finally, Table 2 still presents n=50 (RBP) and n=50 (CON), while all the other tables present n=52/n=50. As mentioned before, the authors should explain such a discrepancy.

Author Response

Response to Reviewer 2

Dear Authors,

Thank you for revising your manuscript. I have carefully read all of your answers, which, for most, are satisfactory.

A few more alterations are needed before publication.

First, the English grammar should be corrected throughout. The editors at IJERPH will hopefully help you in doing so during the proofing of your manuscript.

Response:

Thank you for your suggestion. We try our best to correct the English grammar in the whole manuscript.

Second, regarding my comments on animal welfare, you cited the Declaration of Helsinki, which concerns for 99% Human welfare. Although animal welfare is, indeed, mentioned in the Helskinky Declaration, it is not sufficient to mention such a protocol since animal welfare is not the purpose of the Helsinki declaration.

Response:

Thank you for your inputs. We provide addition sentence highlighting that: “align with the beneficence principle, the use of catfish in the study was beneficial towards acquiring new knowledge and evidence and furthermore the fish was commonly eaten in the site of the study hence benefitting the nutrient intake of the children.”

Moreover, you have stated that information regarding points 2a-2e are included in line 238, which I cannot find. Moreover, point 2f is still not answered. Please include these details in the manuscript.

Response:

Thank you for your inputs. We include the details of points 2a-2f in the manuscript in line 252-257”. “Fish that acquired in one RBP pool was farmed catfish as much as 5000 seeds in one raised bed pool of 2x3 meter. The pool was built in the school yard provided with rooftop for shades. At the end of the study some of the fish were harvested and some were continued to be raised by schools’ guard/personnel. Moreover, stationeries provided each participant after consenting to involve the whole study; the same goes for control group.”

Third, regarding Figure 1, you have now added reference number 14, which is the original paper by Ajzen and Fishbein on planned behaviour. However, your Figure 1 is an adaptation from Figure 5.3 of Ajzen and Fishbein made by Robert Orzanna, under a CC-BY-SA-4.0 licence, which grants free permission to re-use, provided that you cite the authors. Therefore, the caption should read something like "adapted from [14] by Robert Orzanna". Please revise.

Response:

Thank you for your suggestion. We revised Figure 1 as per your suggestion.

Finally, Table 2 still presents n=50 (RBP) and n=50 (CON), while all the other tables present n=52/n=50. As mentioned before, the authors should explain such a discrepancy.

Response:

Thank you for your criticism. We did re-check and re-run analysis on the dataset and found that information on the households’ characteristics of 2 children in control group were missing. However, since they still complete the study, we still used them in the final analysis. We add explanation about these missing data in line 286-289.